# Genome-Wide Identification and Characterization of Long Non-Coding RNAs in Roots of Rice Seedlings under Nitrogen Deficiency

**DOI:** 10.3390/plants12234047

**Published:** 2023-11-30

**Authors:** Dongfeng Qiu, Yan Wu, Kuaifei Xia, Mingyong Zhang, Zaijun Zhang, Zhihong Tian

**Affiliations:** 1Engineering Research Center of Ecology and Agricultural Use of Wetland, Ministry of Education, Hubei Key Laboratory of Waterlogging Disaster and Agricultural Use of Wetland, College of Life Science, Yangtze University, Jingzhou 434025, China; qdflcp@163.com; 2Key Laboratory of Crop Molecular Breeding, Ministry of Agriculture and Rural Affairs, Hubei Key Laboratory of Food Crop Germplasm and Genetic Improvement, Institute of Food Crops, Hubei Academy of Agricultural Sciences, Wuhan 430064, China; yanwu@hbaas.com; 3Key Laboratory of South China Agricultural Plant Molecular Analysis and Genetic Improvement, Guangdong Provincial Key Laboratory of Applied Botany, South China Botanical Garden, Chinese Academy of Sciences, Guangzhou 510650, China; xiakuaifei@scbg.ac.cn (K.X.); zhangmy@scbg.ac.cn (M.Z.)

**Keywords:** long non-coding RNA, nitrogen, rice, ssRNA-Seq, transcriptome

## Abstract

Long non-coding RNAs (lncRNAs) regulate gene expression in eukaryotic organisms. Research suggests that lncRNAs may be involved in the regulation of nitrogen use efficiency in plants. In this study, we identified 1628 lncRNAs based on the transcriptomic sequencing of rice roots under low-nitrogen (LN) treatment through the implementation of an integrated bioinformatics pipeline. After 4 h of LN treatment, 50 lncRNAs and 373 mRNAs were significantly upregulated, and 17 lncRNAs and 578 mRNAs were significantly downregulated. After 48 h LN treatment, 43 lncRNAs and 536 mRNAs were significantly upregulated, and 42 lncRNAs and 947 mRNAs were significantly downregulated. Moreover, the interaction network among the identified lncRNAs and mRNAs was investigated and one of the LN-induced lncRNAs (*lncRNA24320.6*) was further characterized. *lncRNA24320.6* was demonstrated to positively regulate the expression of a flavonoid 3′-hydroxylase 5 gene (*OsF3*′*H5*). The overexpression of *lncRNA24320.6* was shown to improve nitrogen absorption and promote growth in rice seedlings under LN conditions. Our results provide valuable insights into the roles of lncRNAs in the rice response to nitrogen starvation.

## 1. Introduction

Rice is the main source of calories for more than 3.5 billion people [1]. Between 2020 and 2035, the global demand for rice is expected to rise from 763 million tons to 852 million tons, necessitating increased production. Nitrogen (N) is a crucial macronutrient for plants, and its availability is a major factor in determining plant growth and production [2]. In fact, N availability is often the major factor limiting crop yields in nutrient-poor soils [3]. The application of N fertilizer has significantly enhanced agricultural yields and helped global food security, but it has also led to significant environmental issues such as soil acidity and water eutrophication. It is therefore vitally important to improve crop nitrogen-use efficiency (NUE) in order to achieve high yields with low N inputs. Unfortunately, modern breeding programs have often focused on obtaining high yields with high N inputs, resulting in the prevalence of low-NUE cultivars [4]. There have been a considerable number of phenotypic and physiological studies of the plant response to low soil N, but many fewer studies have focused on the associated signaling pathways, molecular mechanisms, and genetic regulation.

Nitrogen deficiency leads to global changes in RNA expression, and genome-wide transcriptomic analyses have been used to broadly identify a variety of stress-responsive noncoding RNAs (ncRNAs) [5]. Long noncoding RNAs (lncRNAs) are defined as noncoding RNAs greater than 200 nucleotides (nt) in length but lacking protein-coding ability [6,7]. lncRNAs can be further subdivided into long intronic ncRNAs (incRNAs) and long intergenic ncRNAs (lincRNAs) based on their genomic position. Although many lncRNAs have been identified, few have been functionally characterized due to their low expression and poor sequence conservation [8]. However, lncRNAs have been found to affect plant development as well as abiotic and biotic stress response [9,10]. For example, the lncRNA *IPS1* is responsive to N and phosphorus (P) starvation [11]. lncRNAs act through an array of mechanisms, including by regulating gene transcription and expression at both the epigenetic and transcriptional levels. Specifically, lncRNAs exhibit cis effects by regulating the expression of local and downstream genes by modulating chromatin conformation, as well as trans effects by regulating the expression of distant genes by interacting with RNA, DNA, or proteins [10,12]. Finally, lncRNAs can act as endogenous target mimics (eTMs) of miRNA to inhibit miRNA accumulation in the cytoplasm, thereby enhancing the expression of miRNA targets at the post-transcriptional level [13,14].

Here, a genome-wide analysis was conducted to detect and identify lncRNAs in rice roots under low N stress. In total, 1638 lncRNA candidates were identified, including 181 (92 repressed and 89 induced) lncRNAs specifically responsive to low N stress. The interaction network between lncRNAs, miRNAs, and mRNAs was also investigated. The lncRNA *lncRNA24320.6* was demonstrated to regulate the expression of a putative flavonoid 3′-hydroxylase 5 gene (*OsF3*′*H5*, LOC_Os09g08920). The overexpression of *lncRNA24320.6* was found to improve N absorption and growth in N-deficient rice seedlings. Overall, our investigation revealed that lncRNAs are important for rice’s reaction to low-nitrogen stress. This finding will encourage future study into the molecular mechanisms of lncRNAs and associated miRNA pathways that underlie rice’s reaction to nutritional deficiency.

## 2. Results

### 2.1. Genome-Wide Identification of lncRNAs in Rice

Strand-specific RNA-seq (ssRNA-Seq) was used to systematically identify lncRNAs responsive to N starvation in rice roots. Specifically, 2-week-old rice seedlings were subjected to N-deficient conditions and their roots were harvested for ssRNA-Seq (Figure 1A). In total, nine samples were generated, producing 42,068,756–59,595,776 clean reads (12.62–17.88 GB clean bases) with a Q30 of 92.07–93.36% and similar GC contents (50.10–52.69%) (Appendix A). The overall alignment efficiency between reads of each sample and the reference genome ranged from 69.67% to 88.32% (Appendix A). These results indicate that our RNA-seq data was of sufficiently high quality for subsequent analyses.

Based on the alignment results, variable splicing prediction analysis, and gene structure optimization analysis, 32,767 genes were identified, including 4190 new genes (Figure 1B). Following the combined CNCI (Coding–Non-Coding Index), CPC (Coding Potential Calculator), Pfam (Protein family) and CPAT (Coding Potential Assessment Tool) analysis, 1638 lncRNAs were identified (Figure 2A and Appendix A). Of these lncRNAs, intergenic lncRNAs (lincRNAs) were the most prevalent (81.6%), followed by anti-sense lncRNAs (9.7%), intronic lncRNAs (6.8%), and sense lncRNAs (2%) (Figure 2B).

To further compare differences in sequence structure between lncRNAs and mRNAs, we analyzed their transcript length, exon number, and open reading frame (ORF) length (Figure 2C–E). Our findings suggest that lncRNAs generally exhibit shorter sequence lengths (ranging from 200 to 1000 bp), whereas mRNA sequences are longer (ranging up to 3000 bp) (Figure 2C). In addition, the majority of lncRNAs consist of two or three exons, whereas mRNAs exhibit a wider distribution of exon numbers (Figure 2D). More than 90% of lncRNAs had an ORF length of ≤200 nt, while 40% of mRNAs had an ORF length of ≥200 nt (Figure 2E). Furthermore, mRNAs were more highly expressed (FPKM = 20.32) on average than lncRNAs (FPKM = 15.06) (Figure 2F). Interestingly, we also found that mRNAs generally have many more isoforms than lncRNAs (Figure 2G).

### 2.2. Identification and Functional Annotation of Differentially Expressed lncRNAs Involved in Nitrogen Starvation

There were more DE mRNAs and lncRNAs after 48 h of N starvation than after 4 h of N starvation (Figure 3A–C). After 4 h of N starvation, 50 lncRNAs and 373 mRNAs were significantly upregulated, while 17 lncRNAs and 578 mRNAs were downregulated (Figure 3A). After 48 h of N starvation, 43 lncRNAs and 536 mRNAs were significantly upregulated, while 42 lncRNAs and 947 mRNAs were downregulated (Figure 3B and Appendix A).

We further validated our results by randomly selecting eight DE lncRNAs and eight DE mRNAs for RT-qPCR analysis. Among these, three lncRNAs and four mRNAs exhibited upregulation after 4 h of N starvation, while five lncRNAs and four mRNAs exhibited downregulation. Overall, these expression levels were consistent with the sequencing results (Appendix A). Eighteen DE lncRNAs were consistent between 4 h and 48 h of N starvation (Figure 3C). The data were consistent with the sequencing results, which confirmed the results were reliable and could be used for intensive studies.

The biological functions of the detected lncRNAs were predicted in both *cis*- and *trans*-acting modes (Appendix A). Co-expression analysis was utilized to predict *trans*-regulated genes based on lncRNA expression levels. Proximal protein-coding genes located within a 100-kb lncRNA genomic window (co-localization) were defined as target genes for *cis* activity (Appendix A).

The GO enrichment analysis of *cis*- and *trans*-regulated genes showed that most target genes were related to starch and sucrose metabolism, carbon metabolism, and the MARP signaling pathway (Figure 3D,E). The KEGG pathway analysis of regulated target genes revealed that lncRNAs were involved in 120 pathways, with the majority related to biotic stress (plant–pathogen interaction), environmental information processing (plant signal transduction), and metabolism (starch and sucrose metabolism, phenylpropanoid biosynthesis) (Figure 3F,G, Appendix A).

### 2.3. Expression of lncRNA24320.6 Is Induced by Nitrogen Deficiency

To evaluate the functions of lncRNAs during N starvation in rice, one of the DE lncRNAs (*lncRNA24320.6*) was further characterized. According to the RNAseq data, *lncRNA24320.6* expression was upregulated after 4 and 48 h of N starvation (Appendix A). *lncRNA24320.6* is predicted to originate from the 5′ UTR and 3′ UTR of LOC_Os09g08910 (ATP synthase F1 homolog), with a full length of 1226 bp (Figure 4A). According to the CPC database, *lncRNA24320.6* had a coding potential score of 0.0108102, indicating that it is non-coding. In contrast, the coding potential score of LOC_Os09g08910 was 1 (Table 1).

The expression of *lncRNA24320.6*, as well as upstream and downstream genes within a 10 kb range, was analyzed in *lncRNA24320.6*-overexpressing transgenic rice plants (*lncRNA24320.6*-*ox*) (Figure 4B,C). The results showed that the expression of LOC_Os09g08920 (a putative flavonoid 3′-hydroxylase 5 gene, *OsF3*′*H5*), located 6 kb downstream of *lncRNA24320.6*, also significantly upregulated both, according to RNAseq and in *lncRNA24320.6-ox* plants (Figure 4C, Appendix A). However, the expression levels of other genes did not change significantly, which is consistent with the RNAseq results (Figure 4C and Appendix A and Appendix A). Under normal conditions, the expression levels of *lncRNA24320.6* and *OsF3*′*H5* were similarly high in roots, leaf sheaths, and panicles, and very low in leaf blades (Figure 5A,B). These results were consistent with the GUS staining results of *lncRNA24320.6* pro::*GUS* plants (Appendix A). The expression of *lncRNA24320.6* and *OsF3*′*H5* was also similarly induced by N deficiency (Figure 5C,D). These results suggest that *lncRNA24320.6* positively regulates the expression of *OsF3*′*H5* under N deficiency in rice.

### 2.4. lncRNA24320.6 Promotes Nitrogen Utilization under Low Nitrogen Conditions

To evaluate the role of *lncRNA24320.6* in the rice N starvation response, *lncRNA24320.6-ox* plants were grown hydroponically under various N conditions. After two weeks of low N treatment, the N and C contents of roots and shoots were measured (Figure 6). Under normal N conditions, both *lncRNA24320.6-ox* and ‘ZH11′ plants exhibited similar growth patterns. After two weeks of low N (0N and 1/4N) treatment, the roots and shoots of both *lncRNA24320.6-ox* and ‘ZH11′ plants contained similar C content (Figure 6A,B). However, the *lncRNA24320.6-ox* plants contained significantly more N in roots and shoots than ‘ZH11′ plants (Figure 6C,D). Interestingly, under high N conditions (2N), the roots of *lncRNA24320.6-ox* and ‘ZH11′ plants exhibited similar N contents, but the shoots of *lncRNA24320.6*-*ox* plants exhibited a lower N content than the shoots of ‘ZH11′ plants (Figure 6C,D). These results indicated that *lncRNA24320.6* can regulate the accumulation of nitrogen in rice roots.

The growth and agronomic characteristics of *lncRNA24320.6-ox* plants and ZH11 plants were evaluated in a controlled field with and without N fertilizer application (Figure 7). Under normal conditions (CK), *lncRNA24320.6-ox* plants were taller and had more tillers than ‘ZH11′ plants (Figure 7A,B), although the plants did not differ in either grain weight or seed setting rate (Figure 7C,D). Under low N conditions (LN), the plants did not differ in height (Figure 7A), but *lncRNA24320.6-ox* plants had more tillers, a higher grain weight, and a higher seed setting rate than ‘ZH11′ plants (Figure 7B–D). This result indicates that the overexpression of *lncRNA24320.6* will not cause harm to rice growth, even improve both the growth and grain yield of rice under low N conditions.

## 3. Discussion

Considerable evidence indicates that lncRNAs are essential regulators of various biological processes in plants, including nutrient response, growth and development, and response to biotic and abiotic stresses [12]. In this study, a total of 181 DE lncRNAs were screened in rice roots under N deficiency. One of them (*lncRNA24320.6*) was found to regulate the expression of *OsF3*′*H5* (LOC_Os09g08920) in response to N starvation.

### 3.1. Enrichment of lncRNAs in Response to Low Nitrogen Stress in Rice Roots

lncRNAs are known to play important roles in the plant response to stress, such as high salinity or pathogen infection [15,16]. Whole genome RNA sequencing (RNA-Seq) has been instrumental in the genome-wide identification of lncRNAs, including leading to the identification of 4373 lncRNAs in *Arabidopsis thaliana*, approximately 12,000 lncRNAs in *Brassica napus*, 10,761 lncRNAs in *Zea mays* (CANTATAdb2.0, http://cantata.amu.edu.pl/, 27 November 2023), and 2788 lncRNAs in *O. sativa* [17,18,19,20]. To date, several lncRNAs have also been functionally characterized. For example, in *Arabidopsis* a lncRNA *ASCO* (alternative splicing competitor) has been proven to bind to alternative splicing and play a crucial regulatory role in root development [21]. An antisense lncRNA, TCONS-00000966, was responsive to *Sclerotinia sclerotiorum* infection in *Brassica napus* [22]. The 1236 bp photoperiod-sensitive long-day-specific male fertility-associated RNA (*LDMAR*) controls male sterility in rice, and a sufficient amount of the *LDMAR* transcript is necessary for proper pollen formation under long-day circumstances [23].

In this study, 1638 lncRNAs were identified from the roots of rice seedlings (Figure 2A, Appendix A). The lncRNAs generally exhibited shorter sequence lengths (ranging from 200 to 1000 bp) and fewer exons than mRNAs (Figure 2C,D). These features are generally consistent with previous reports, such as those on rice brown planthopper (*Nilaparvata lugens*) and silkworm [24,25]. The number of lncRNAs identified was less than previously reported for rice, likely due to the use of only nine N-deficient root samples. We identified 50 lncRNAs and 373 mRNAs which were significantly upregulated after 4 h of N starvation, as well as 17 lncRNAs and 578 mRNAs, which were downregulated (Figure 3A). After 48 h of N starvation, 43 lncRNAs and 536 mRNAs were significantly upregulated, while 42 lncRNAs and 947 mRNAs were downregulated (Figure 3B). Our findings may provide an important lncRNA database for further exploring the regulation of NUE via lncRNAs in rice.

### 3.2. A Few Pairs of DE lncRNAs and mRNAs May Influence NUE of Rice Roots

Co-expression network analysis can uncover lncRNA and mRNA interactions, which is essential to functionally characterize lncRNAs. lncRNAs enact the *cis*-regulation of their neighboring protein coding genes [26]. In our study, we identified 1638 co-expressed lncRNA and mRNA transcripts (Appendix A), including a few genes known to influence NUE. For example, *AMT3.1* (Os02g0550800), *OsPTR17* (Os01g0902700), and *NRT2.4* (Os01g0547600) were upregulated in response to N deficiency, consistent with the results of Shin. [23]. Based on these data, we identified two pairs of DE lncRNAs and mRNAs, which may influence the NUE of rice roots. lncRNA MSTRG.12144.19 was able to positively regulate the expression of *OsNRT2.3a* (LOC_Os01g50820), which plays a key role in long-distance nitrate transport from root to shoot at low nitrate supply levels in rice [27]. Another lncRNA, MSTRG.4764.4, targets a member of the plant NITRATE TRANSPORTER 1/PEPTIDE TRANSPORTER (NRT1/PTR) family (LOC4338779), and existing research has proved that these plant proteins transport a wide variety of substrates: nitrate, peptides, amino acids, dicarboxylates, glucosinolates, IAA, and ABA [28].

The identification of 181 N deficiency-responsive DE lncRNA genes in rice roots suggests that lncRNAs might play key roles in the rice response to N starvation (Appendix A). Our results indicated that *lncRNA24320.6* is inducible under low N conditions and plays an important role in N absorption. *lncRNA24320.6* is predicted to originate from the 5′ UTR and 3′ UTR of LOC_Os09g08910 (ATP synthase F1 homolog), with a full length of 1226 bp (Figure 4A). The overexpression of *lncRNA24320.6* improved both the growth and grain yield of rice under low N conditions (Figure 6 and Figure 7), and *lncRNA24320.6* may positively regulate the expression of *OsF3*′*H5* (LOC_Os09g08920) (Figure 4 and Figure 5). *OsF3*′*H5* is predicted to encode a putative flavonoid 3′-hydroxylase 5, which is an important enzyme in determining the B-ring hydroxylation pattern of flavonoids in rice [29]. Two flavonoid 3′-hydroxylase (CYP93G1 and CYP93G2) are involved in tricin biosynthesis and affects lignification [30]. In rice leaves, nitrate deficiency can increase lignin biosynthesis and flavonoid metabolism [31]. Therefore, the *lncRNA24320.6-OsF3*′*H5* regulatory module may affect rice leaf lignification under N deficiency, so that normal plant growth can be maintained under low nitrogen conditions.

## 4. Conclusions

Long non-coding RNAs (lncRNAs) play an important role in the regulation of nitrogen use efficiency in plants. In this study, we identified lncRNAs and miRNAs based on the transcriptomic sequencing of rice roots under low-nitrogen (LN) treatment. Moreover, the interaction network among the identified lncRNAs and mRNAs was investigated and one of the LN-induced lncRNAs (*lncRNA24320.6*) was further characterized. *lncRNA24320.6* was demonstrated to positively regulate the expression of a flavonoid 3′-hydroxylase 5 gene (*OsF3*′*H5*). The overexpression of *lncRNA24320.6* was shown to improve nitrogen absorption and promote growth in rice seedlings under LN conditions. Our results provide valuable insights into the roles of lncRNAs in the rice response to nitrogen starvation.

## 5. Materials and Methods

### 5.1. Plant Materials and Low-Nitrogen Treatment

The seeds of ZH11 rice were soaked and germination at 37 °C. The seeds with consistent germination were grown in a solution under hydroponic conditions for four weeks [32]. The solution was matched with deionized water and refreshed every 3 days. Four-week-old seedlings were transferred to N-deficient (LN, 0.2 mM NH_4_NO_3_) or N-sufficient (NN, 1.44 mM NH_4_NO_3_) nutrients, and the roots were harvested for RNA extraction and sequencing separately at 0, 4, and 48 h after transfer (Figure 1A). The samples were immediately frozen in liquid nitrogen and stocked in −80 °C until they were used. Three independent biological replicates were used.

### 5.2. RNA Extraction

Total RNA was isolated using a RNAiso Plus kit (TaKaRa, Takara Bio Inc., Dalian, Liaoning, China), according to the manufacturer’s instructions. RNA degradation and contamination were checked using 1% agarose gels. RNA concentration and purity were evaluated using a NanoDrop 2000 Spectrophotometer (Thermo Fisher Scientific, Waltham, MA, USA).

### 5.3. Library Preparation

Library preparation and sequencing were performed by Biomarker Technologies Co. Ltd. (Beijing, China). Sequencing libraries were generated using an NEB Next Ultra Directional RNA Library Prep Kit for Illumina (New England Biolabs, Beijing, China), with index codes added to attribute sequences to each sample. Library quality was assessed using an Agilent Bioanalyzer 2100 (Agilent, Berlin, Germany) and qPCR(Roche, Basel, Switzerland).

### 5.4. Clustering and Sequencing

The clustering of the index-coded samples was performed on a cBot Cluster Generation System using a TruSeq PE Cluster Kitv3-cBot-HS (Illumina, San Diego, CA, USA, https://www.illumina.com.cn/, 27 November 2023). Following cluster generation, library preparations were sequenced on an Illumina platform and reads were generated.

### 5.5. Quality Control

Raw data in fastq format were first processed using in-house keep scripts. During this step, clean data were obtained by removing reads containing adapters and poly-N, as well as low-quality reads. The Q20, Q30, GC-content, and sequence duplication level of the clean data were also calculated. All downstream analyses were based on high-quality clean data.

### 5.6. Identification of lncRNAs and mRNAs

The transcriptome was assembled from clean reads using StringTie [33]. The *Oryza sativa* reference genome from NCBI was used for sequence alignment and subsequent analyses. Gffcompare program was used to annotate assembled transcripts. The reads with either a perfect match or one mismatch were analyzed and annotated. A flowchart of the HISAT2 StringTie analysis is shown in Appendix A. Four computational approaches, CPC (Coding Potential Calculator), CNCI (Coding–Non-Coding Index), Pfam (Protein family) and CPAT (Coding Potential Assessment Tool) were combined to screen for putative lncRNAs,. Transcripts with lengths greater than 200 nt and with more than two exons were selected as candidate lncRNAs [34,35]. All lncRNAs were distinguished using cuffcompare.

### 5.7. Identification of Differentially Expressed lncRNAs and mRNAs

lncRNAs and mRNAs expression was calculated according to the Fragments Per Kilobase of transcript per Million fragments mapped (FPKM) by StringTie (1.3.1) [36]. Differential expression (DE) analysis was performed using DESeq and edgeR [37,38]. Genes with an absolute value of log_2_ (fold change) ≥ 1 and an adjusted *p*-value < 0.05 were defined as DE. The *p*-value was calculated using the Benjamini and Hochberg approach.

### 5.8. Target Gene Prediction and Functional Enrichment Analysis

The predicted target genes of the lncRNAs were classified into *cis*- and *trans*-acting groups based on their regulatory patterns [39]. Perl script was used to identify potential *cis* target genes with the threshold of 100 kb upstream or downstream of lncRNAs [40]. In addition, *trans* target genes were identified by measuring the correlation between the expression levels of lncRNAs and mRNAs. Pearson’s correlation coefficient was used to evaluate the relevance of expression (|r| > 0.9 and *p* < 0.01) [40]. To functionally characterize target genes, we performed Gene Ontology (GO) and Kyoto Encyclopedia of Genes and Genomes (KEGG) pathway analyses by referencing the GO database (http://geneontology.org/, 27 November 2023) and KEGG pathway database (https://www.kegg.jp/, 27 November 2023), respectively.

### 5.9. Validation by Quantitative Real-Time PCR (qRT-PCR)

Total RNA was collected from the roots of rice seedlings with RNAiso plus kit (Takara, Takara Bio Inc., Dalian, China). Reverse transcription and qRT-PCR amplification were performed as previously described [41]. The rice *e-EF-1α* gene was used as an internal control, and the expression of differentially expressed lncRNAs were calculated using the 2^−ΔΔCt^ method [42]. Each experiment consisted of three biological and three technical replicates. All primers are listed in Appendix A.

### 5.10. Construction of Transgenic Rice Overexpressing lncRNA24320.6 and Pro-lncRNA24320.6::GUS

To generate the *lncRNA24320.6* overexpression vector (*lnc24320.6-ox*), its full-length cDNA was amplified from leaves of ‘ZH11′ rice and was subcloned into the pXU1301 vector by using *Hind*III and *Bam*HI sites downstream of the *Ubi* promoter. To generate the pro*-lncRNA24320.6::GUS* vector, approximately 2000 bp of genomic DNA sequence upstream of *lncRNA2432.6* was subcloned into pCambia1301 using the *Afl*II and *Bgl*II restriction sites. These vectors were introduced into *Agrobacterium tumefaciens* strain *‘EHA105*′, and then ‘ZH11′ rice was subjected to *Agrobacterium*-mediated transformation using 50 mg/L hygromycin to select transgenic calli. More than ten individual homozygous or heterozygous plants were obtained, and two homozygous lines were selected for further experiments.

### 5.11. Phenotypic Evaluation

The homozygote generation of both transgenic (T_3_ and T_4_) and control rice plants (ZH11) were planted in the field at South China Botanical Garden, CAS, Xingke Road, Tianhe District, Guangzhou, China. The field experiment followed a randomized block design with two N levels (+N, 150 kg/ha N fertilizer; 1/2 N, 75 kg/ha N fertilizer). Under both conditions, both P and potassium (K) fertilizers were applied at a rate of 100 kg/ha. Total P and K, and 60% of N, fertilizers were applied by top-dressing prior to transplantation, and the remaining 40% of N was applied at the tillering stage. At least twenty-four plants of each line were used for phenotype analysis. Three biological replications were used, and every plant was placed randomly. Plants were spaced 13.3 cm apart, and rows were 26.6 cm apart when seedlings were planted in the field. Throughout the whole growth phase, field management was carried out according to standard agronomic practices.

We evaluated root and shoot length, total N content, and total carbon (C) content under N-deficient (LN, 0.24 mM NH_4_NO_3_), N-sufficient (NN, 1.44 mM NH_4_NO_3_), and N-excessive (2N, 2.88 mM NH_4_NO_3_) hydroponic conditions, as well as number of tillers, length of panicles, and 1000-seed-weight under +N and 1/2 N field conditions.

## Figures and Tables

**Figure 1 plants-12-04047-f001:**
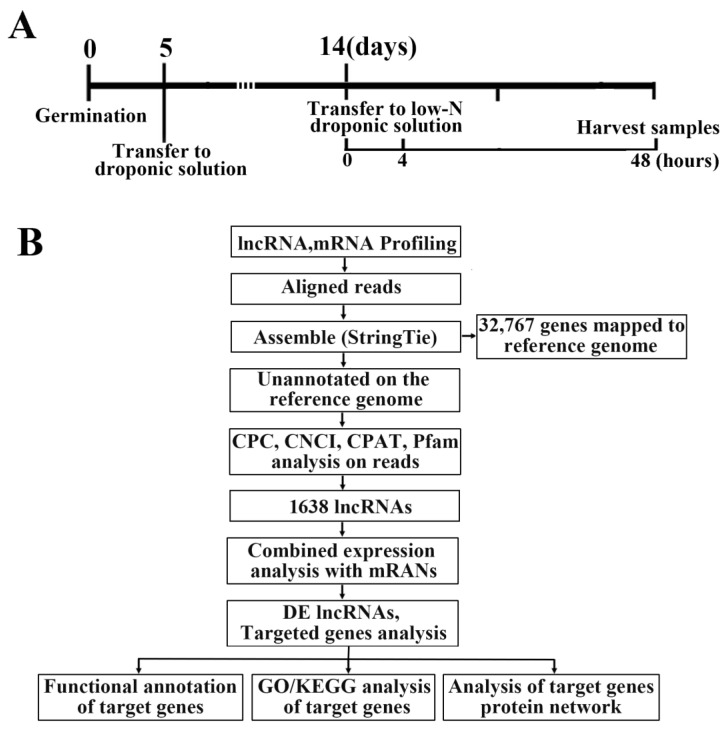
Identification of differentially expressed (DE) lncRNAs in rice roots under low nitrogen treatment. (**A**) Preparation of nitrogen-starved ‘Zhonghua 11’ (ZH11) rice seedlings for RNA sequencing. The 14-day-old rice roots were subjected to nitrogen deficiency for 4 h and 48 h prior to sampling. (**B**) Screening of total and DE lncRNAs.

**Figure 2 plants-12-04047-f002:**
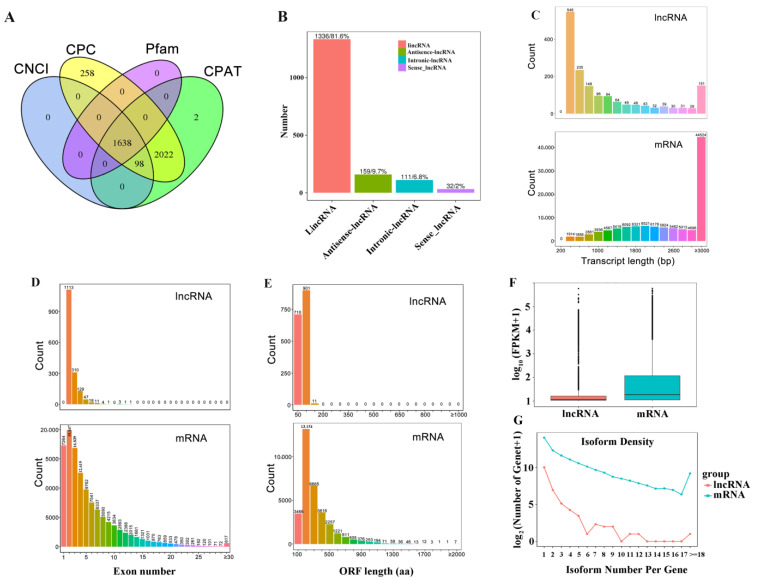
Identification and characterization of lncRNAs in rice seedlings. (**A**) Venn diagram of putative lncRNAs screened via coding potential analysis methods, including CNCI, CPC, Pfam, and CPAT. Numbers within circles represent the number of lncRNA transcripts predicted to be positive. The intersection of the four circles is taken as the prediction result. (**B**) lncRNA composition. lincRNA: long chain non-coding RNA in intergenic region; antisense lncRNA: antisense long chain non-coding RNA; intronic lncRNA: intronic long chain non-coding RNA; sense lncRNA: long strand non-coding RNA. (**C**) Comparison of lncRNA and mRNA transcript lengths. The x-axis is the length, and the y-axis is the number of lncRNAs/mRNAs whose length is distributed within the range. (**D**) Comparison of the number of exons between lncRNAs and mRNAs. (**E**) Comparison of open reading frame (ORF) lengths between lncRNAs and mRNAs. (**F**) Comparison of lncRNA and mRNA expression levels. The bottom of the box chart is the maximum, the upper quartile, the median, the lower quartile, and the minimum. (**G**) Comparison of isoform numbers per gene between lncRNAs and mRNAs. The x-axis represents the distribution of the number of variable shear isomers per gene, and the y-axis represents log_2_ (number of genes+1).

**Figure 3 plants-12-04047-f003:**
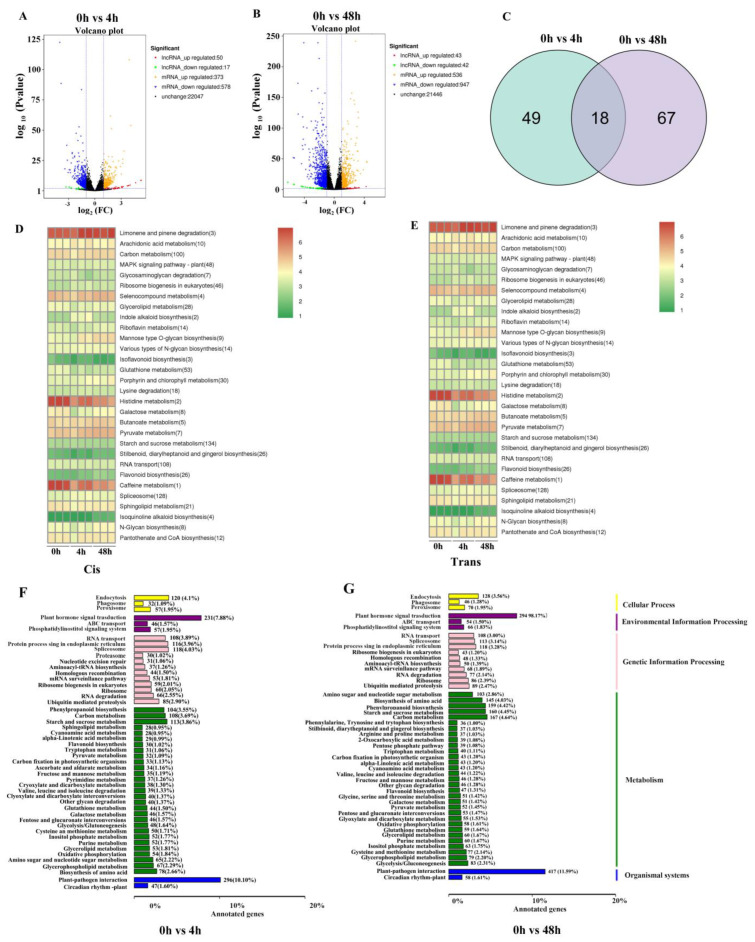
Differentially expressed (DE) genes and functional annotation involved in nitrogen deficiency. (**A**) Volcano plot of DE mRNAs and lncRNAs 4 h after nitrogen starvation. (**B**) Volcano plot of DE mRNAs and lncRNAs 48 h after nitrogen starvation. (**C**) Venn diagram of DE lncRNAs 4 h and 48 h after nitrogen starvation. (**D**) GO enrichment analysis of *cis*-regulated target genes. (**E**) GO enrichment analysis of *trans*-regulated target genes. Red represents high enrichment functional classification, and blue represents terms with relatively low enrichment. The numbers in each term label indicate the number of DE genes. (**F**) KEGG annotation of DE lncRNA-targeted genes 4 h after nitrogen starvation. (**G**) KEGG annotation of DE lncRNA-targeted genes 48 h after nitrogen starvation. The vertical axis represents the name of the KEGG metabolic pathway, and the horizontal axis represents the number of lncRNA-targeted genes annotated to this pathway and their proportion to the total number of lncRNA-targeted genes annotated.

**Figure 4 plants-12-04047-f004:**
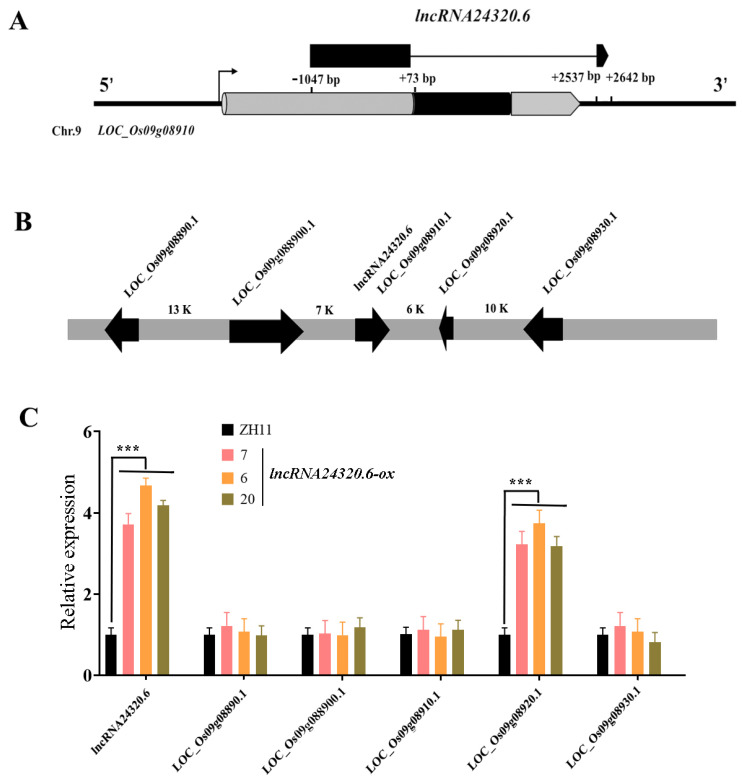
Location and targeted gene prediction of *lncRNA24320.6*. (**A**) Location of *lncRNA24320.6*. (**B**) Predicted genes *cis*-targeted by *lncRNA24320.6*. (**C**) Relative expression of *lncRNA24320.6* and its predicted *cis*-targeted genes in *lncRNA-24320.6-ox* plants. The housekeeping gene *eEF-1α* (Os03g0178000) was used as a reference gene to normalize the relative expression. Three biological and three technical replicates were used for each experiment. *** represents *p*-value < 0.001.

**Figure 5 plants-12-04047-f005:**
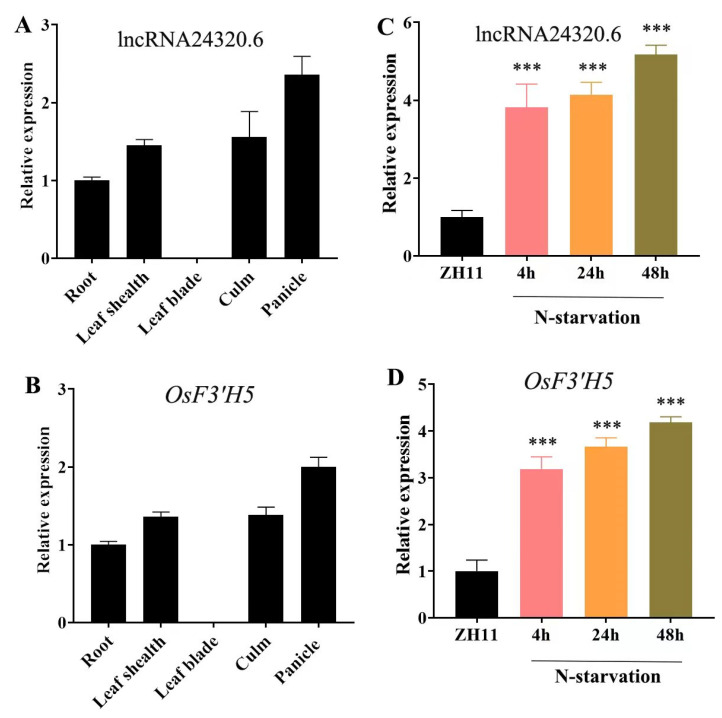
Expression patterns of *lncRNA24320.6* and *OsF3*′*H5*. (**A**) Relative expression of *lncRNA24320.6* in different tissues of heading rice plants. (**B**) Relative expression of *OsF3*′*H5* in different tissues of heading rice plants. (**C**) Relative expression of *lncRNA24320.6* in nitrogen-starved rice seedlings. (**D**) Relative expression of *OsF3*′*H5* in nitrogen-starved rice seedlings. *** represents *p*-value < 0.001.

**Figure 6 plants-12-04047-f006:**
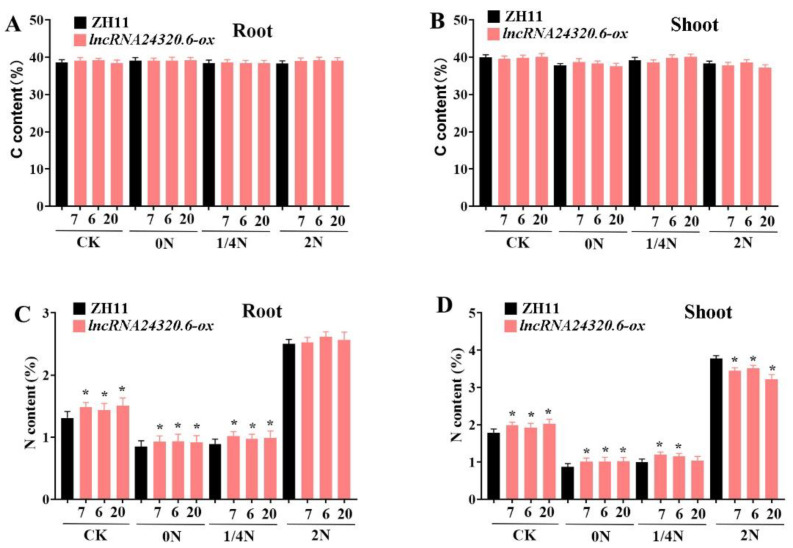
Total carbon and nitrogen contents in *lncRNA24320.6-ox* and ‘ZH11′ rice under various nitrogen conditions. (**A**) Total carbon content in roots. (**B**) Total carbon content in shoots. (**C**) Total nitrogen content in roots. (**D**) Total nitrogen content in shoots. * represents *p*-value < 0.05.

**Figure 7 plants-12-04047-f007:**
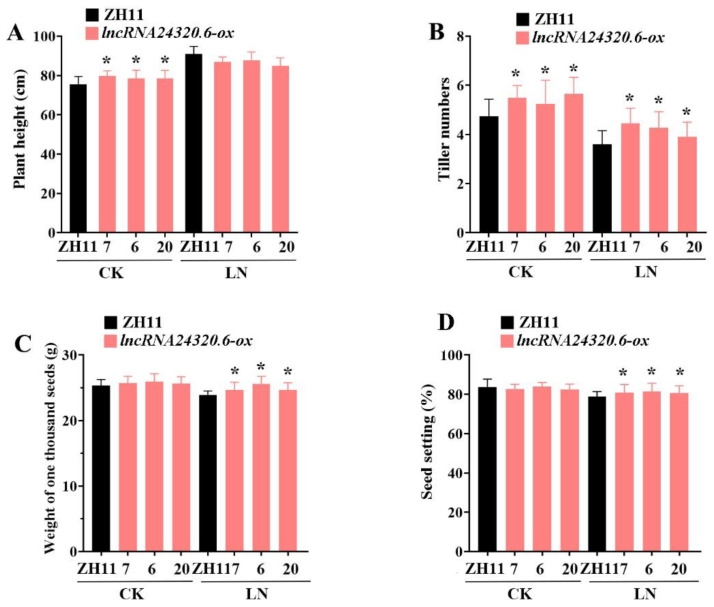
Agronomic traits of *lncRNA24320.6-ox* and ‘ZH11′ rice under normal (CK) and low nitrogen (LN) field conditions. (**A**) Plant height. (**B**) Tiller number. (**C**) Weight of one thousand seeds. (**D**) Seed setting rate. Data are the means of three biological replicates with a standard deviation (n = 30). * represents *p*-value < 0.05.

**Table 1 plants-12-04047-t001:** The coding capability of *lncRNA24320.6***.** LOC_ Os09g08910 was used as a positive control. C/NC indicates whether the coding protein is present or not. The coding potential score represents encoding capability, with a value of 1 indicating encoding ability and a value of less than 1 indicating a lack of encoding ability.

RNA Name	C/NC	Coding Potential Score
*lncRNA24320.6*	Noncoding	0.0108102
LOC_Os09g08910	Coding	1

## Data Availability

The RNA sequencing data were deposited in the SRA database at the NCBI under accession number PRJDB16072. All other data and material analyzed in the current study are included in the manuscript and the Appendix A.

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
