# Peer review of "Genome-Wide Identification and Characterization of Long Non-Coding RNAs in Roots of Rice Seedlings under Nitrogen Deficiency"

_plants, 2023, doi:10.3390/plants12234047_

Round 1

Reviewer 1 Report

Comments and Suggestions for Authors

Arrange keywords in alphabetic order. 

The reference list should be updated with more recent references, little references cited in 2023.

The aim of the work should be improved. 

How many rice cultivars used?

Write in M &M the source of rice grains. 

How many replicates used per treatment?

Please mention in your M&M, how the rice grains were sterilized? Are you used rice seedlings in the hydroponic culture? if yes, what is the age of these seedlings?

I highly recommend rewriting the first paragraph in M&M section, lots of information is lost. 

How the authors preserve the harvested tissues?

Improve the discussion section.

Where is the conclusion part?

Improve the quality of figure 3. 

Improving the expression of results is also needed.  

Comments on the Quality of English Language

The language should be improved.

Author Response

[Response]: We are grateful for your encouraging comments and insightful suggestions. The manuscript has been revised thoroughly and the corrections and re-written portions are marked in text (highlighted contents). Responses to the comments are listed in a point-by-point fashion as follows:

Arrange keywords in alphabetic order.

[Response]: Thank you for your suggestion. Follow your suggestion, we have changed the keywords in alphabetic order in line35, as the order of “long non-coding RNA; nitrogen; rice; ssRNA-Seq; transcriptome”.

The reference list should be updated with more recent references, little references cited in 2023.

[Response]: Thank you for your suggestion. Follow your suggestion, we have updated the references, please see the highlighted contents of “References” in the revised manuscript.

The aim of the work should be improved.

[Response]: Thank you for your suggestion. Follow your suggestion, we have rewritten the significance of the work in the discussion. please see the highlighted contents of “Discussion” in the revised manuscript.

How many rice cultivars used?

[Response]: Thank you for your question. For phenotypic evaluation, at least 24 plants of each line and three biological replications were used which was showed in section 5.11 in the revised manuscript. Please see the highlighted contents of “Materials and methods” in the revised manuscript.

Write in M &M the source of rice grains.

[Response]: Thank you for your suggestion. Follow your suggestion, we have written the source of rice grains in the line 316. Please see the highlighted contents of “Materials and methods” in the revised manuscript.

How many replicates used per treatment?

[Response]: Thank you for your question. Three independent biological replicates were used in this research. And it was shown in lines 398-399 in the revised manuscript. Please see the highlighted contents of “Materials and methods” in the revised manuscript.

Please mention in your M&M, how the rice grains were sterilized? Are you used rice seedlings in the hydroponic culture? if yes, what is the age of these seedlings?

[Response]: Thank you for your question. We have not sterilized the rice grains, and all the seeds were common germinated at 37oC. The rice seedlings are in the hydroponic culture, and four-week-old seedling were used to further treatments. Also, we add the illustration about the materials and age in section 5.1 (lines 316-323). Please see the highlighted contents of “Materials and methods” in the revised manuscript.

I highly recommend rewriting the first paragraph in M&M section, lots of information is lost.

[Response]: Thanks for your suggestion. We have rewrite it in detail in section 5.1 (lines 316-323). Please see the highlighted contents of “Materials and methods” in the revised manuscript.

How the authors preserve the harvested tissues?

[Response]: The roots were harvested for RNA extraction and sequencing separately at 0, 4, and 48 h after transfer. The samples were immediately frozen in liquid nitrogen and stocked in -80oC until they were used. The details of the materials were rewrite in section 5.1 (lines 316-323). Please see the highlighted contents of “Materials and methods” in the revised manuscript.

Improve the discussion section.

[Response]: Thank you for your suggestion. Follow your suggestion, we have rewritten the discussion (lines 253-302). please see the highlighted contents of “Discussion” in the revised manuscript.

Where is the conclusion part?

[Response]: Thank you for your reminding, the conclusion was added after discussion, please see the highlighted contents of lines 303-312 in the revised manuscript.

Improve the quality of figure 3.

[Response]: Thank you for your suggestion. we have revised it in new Figure 3 in the revised manuscript.

Improving the expression of results is also needed.

[Response]: Thank you for your suggestion. Follow your suggestion, we have rewritten the results. please see the highlighted contents of “Results” in the revised manuscript.

Reviewer 2 Report

Comments and Suggestions for Authors

all comments are included within the file

Author Response

all comments are included within the file

[Response]: We are grateful for your encouraging comments, thank you very much.

Reviewer 3 Report

Comments and Suggestions for Authors

Congratulations on a job well done

Author Response

Congratulations on a job well done

[Response]: We are grateful for your encouraging comments, thank you very much.

Reviewer 4 Report

Comments and Suggestions for Authors

I checked your manuscript and described comments below.

Rice is an important grain eaten all over the world. This paper performs a very good genome-wide analysis of rice non-coding RNA under nitrogen starvation.

I think you should consider the following points.

1.       How many of the 1,628 lnc RNAs identified in this study were novel? I think it would be better to write in detail.

2.       The sequence information for lncRNA24320.6 is not well understood. I think it would be better to describe where it is located in Supplementary files.

3.       "2.2. A few pairs of DE lncRNAs and mRNAs may influence NUE of rice roots" is difficult to understand, so I think it would be better to create a diagram.

I don't think this paper has new various major mistakes or grammatical problems.

Author Response

I checked your manuscript and described comments below.

Rice is an important grain eaten all over the world. This paper performs a very good genome-wide analysis of rice non-coding RNA under nitrogen starvation.

I think you should consider the following points.

[Response]: We are grateful for your encouraging comments and insightful suggestions. The manuscript has been revised thoroughly and the corrections and re-written portions are marked in text (highlighted contents). Responses to the comments are listed in a point-by-point fashion as follows:

1.How many of the 1,628 lncRNAs identified in this study were novel? I think it would be better to write in detail.

[Response]: Thank you for your question. Our sequencing was finished in 2017, there were few lncRNAs have been identified at that time. So, there are only 4 lncRNAs was identified as the identified lncRNAs, 1624 lncRNAs were classified as novel at that time.

2.The sequence information for lncRNA24320.6 is not well understood. I think it would be better to describe where it is located in Supplementary files.

[Response]: Thanks for your suggestion. I have marked out the location and length of lncRNA24320.6 in Figure 4 to make it more clearly to show its location.

  1. “2.2. A few pairs of DE lncRNAs and mRNAs may influence NUE of rice roots" is difficult to understand, so I think it would be better to create a diagram.

[Response]: Thanks for your suggestion. Through bioinformatics analysis and literature search, we have identified three pairs of DE lncRNAs and mRNAs may influence NUE of rice roots. MSTRG.12144.19-OsNRT2.3a, MSTRG.4764.4-NRT1, and lncRNA24320.6-OsF3'H5. We have indicated these pairs in the discussion, please  see the highlighted contents of “Discussion” lines 281-287 and 291-293 in the revised manuscript.

I don't think this paper has new various major mistakes or grammatical problems.

[Response]: We are grateful for your encouraging comments and insightful suggestions.

Round 2

Reviewer 1 Report

Comments and Suggestions for Authors

Accept in the current format. 

Reviewer 2 Report

Comments and Suggestions for Authors

No issue